# ATP Synthase Subunit Epsilon Overexpression Promotes Metastasis by Modulating AMPK Signaling to Induce Epithelial-to-Mesenchymal Transition and Is a Poor Prognostic Marker in Colorectal Cancer Patients

**DOI:** 10.3390/jcm8071070

**Published:** 2019-07-21

**Authors:** Yan-Jiun Huang, Yi-Hua Jan, Yu-Chan Chang, Hsing-Fang Tsai, Alexander TH Wu, Chi-Long Chen, Michael Hsiao

**Affiliations:** 1Division of Colorectal Surgery, Department of Surgery, Taipei Medical University Hospital, Taipei Medical University, Taipei 110, Taiwan; 2Department of Surgery, College of Medicine, Taipei Medical University, Taipei 110, Taiwan; 3Genomics Research Center, Academia Sinica, Taipei 115, Taiwan; 4The PhD Program for Translational Medicine, College of Medical Science and Technology, Taipei Medical University, Taipei 110, Taiwan; 5Graduate Institute of Medical Sciences, National Defense Medical Center, Taipei 114, Taiwan; 6Department of Pathology, Taipei Medical University Hospital, Taipei Medical University, Taipei 110, Taiwan; 7Department of Pathology, College of Medicine, Taipei Medical University, Taipei 110, Taiwan; 8Department of Biochemistry, College of Medicine, Kaohsiung Medical University, Kaohsiung 807, Taiwan

**Keywords:** ATP5E, AMPK, EMT, Metastasis, Colorectal Cancer

## Abstract

Metastasis remains the major cause of death from colon cancer. We intend to identify differentially expressed genes that are associated with the metastatic process and prognosis in colon cancer. ATP synthase epsilon subunit (*ATP5E*) gene was found to encode the mitochondrial F_0_F_1_ ATP synthase subunit epsilon that was overexpressed in tumor cells compared to their normal counterparts, while other genes encoding the ATP synthase subunit were repressed in public microarray datasets. CRC cells in which ATP5E was silenced showed markedly reduced invasive and migratory abilities. ATP5E inhibition significantly reduced the incidence of distant metastasis in a mouse xenograft model. Mechanistically, increased ATP5E expression resulted in a prominent reduction in E-cadherin and an increase in Snail expression. Our data also showed that an elevated *ATP5E* level in metastatic colon cancer samples was significantly associated with the AMPK-AKT-hypoxia-inducible factor-1α (HIF1α) signaling axis; silencing ATP5E led to the degradation of HIF1α under hypoxia through AMPK-AKT signaling. Our findings suggest that elevated ATP5E expression could serve as a marker of distant metastasis and a poor prognosis in colon cancer, and ATP5E functions via modulating AMPK-AKT-HIF1α signaling.

## 1. Introduction

Colorectal cancer (CRC) is one of the leading causes of morbidity and mortality worldwide, with about 1.4 million new cases reported in 2012 [1]. There was a global increase in CRC mortality, rising from an estimated 608,700 deaths in 2008 to almost 700,000 deaths in 2012 [1,2]. As a major clinical and public health concern, it is the third most commonly diagnosed cancer and the second leading cause of cancer-related deaths in both genders in the United States. Mortality resulting from CRC is associated with the disease stage, more-advanced grade, and the presence of obstruction [3]. Among these prognostic indicators, metastasis to distant organs (e.g., lung metastasis) is one of the most critical causes related to mortality [4]. Approximately 50% of patients develop distant metastases within 2 years after surgery and have a poor prognosis. Identifying a reliable diagnostic marker could serve to improve the management of patients with metastatic CRC.

Distant metastasis is one of the hallmarks of cancer and is comprised of a series of complex and interconnected cellular signaling networks. The so-called epithelial-to-mesenchymal transition (EMT) was shown to be the initiating and prerequisite cellular process for cancer cells to gain the ability to metastasize. The EMT is characterized by cancer cells’ transformation from an epithelial phenotype to a mesenchymal phenotype, often reflected by increased vimentin (a mesenchymal marker) expression with a concomitant decrease in E-cadherin (an epithelial marker) expression [5,6]. Accumulating evidence suggests a close link between initiation of the EMT and metabolic reprogramming within cancer cells [7,8].

Bioenergetic proteins within mitochondria were found to possess the potential to be prognostic markers associated with cancer progression [9,10]. Among the metabolic pathways, adenosine monophosphate-activated protein kinase (AMPK) is considered a crucial energy sensor for regulating and adapting to hypoglycemic states [11,12]. To curb catabolic activity in the setting of energy depletion, phosphorylated (p)-AMPK interferes with Akt signaling through direct inhibition [13]. Furthermore, inhibiting the phosphorylation of Akt activates glycogen synthase kinase-3β (GSK3β) and consequently destabilizes Snail, which induces the expression of E-cadherin [14].

The Warburg effect is when cancer cells mainly generate their energy by glycolysis instead of oxidative phosphorylation; it is utilized by normal cells due to impaired mitochondrial function [15]. In line with Warburg’s hypothesis, reports have shown that the β-catalytic subunit of H+-adenosine triphosphate (ATP) synthase is downregulated in renal and colon carcinomas along with the upregulation of glycolytic glyceraldehyde 3-phosphate dehydrogenase (GAPDH); the metabolic phenotype in these cells is considered a tumor progression marker with prognostic value for early-stage patients [16]. In addition, significant increases in ATP synthase α- and δ-subunits expressions were observed in primary tumors compared to the normal mucosa, while downregulation of the α- and δ-subunits led to decreased invasion in vitro in liver metastasis of primary CRC [17,18]. Based on these premises, we hypothesized that metastasis is an energy-demanding process that prompts cancer cells to acquire extra energy by upregulating energy-producing machinery. Increased ATP and ATP synthase hence appear to be potential targets for cancer therapy [19,20]. As one of the major mitochondrial enzymes, ATP synthase produces ATP and provides energy by driving phosphorylation of adenosine diphosphate (ADP) through a transmembrane proton gradient [21]. The enzyme includes an F0 sector composed of hydrophobic subunits for energy transduction, as well as an F1 sector composed of hydrophilic α-, β-, γ-, δ-, and ε-subunits for its catalytic function [22,23]. As to the roles of ATP synthase subunits in cancer development, little is known about the ε-subunit, which is encoded by the human ATP synthase epsilon subunit (*ATP5E*) gene and is located in the stalk region of the F1 sector [24,25]. A mutation in *ATP5E* leads to an isolated ATP synthase deficiency and mitochondrial disease, while the ε-subunit seems to be linked to incorporation of the c-subunit [26]. Knockdown of *ATP5E* inhibited the biogenesis of ATP synthase, reduced the ATP synthase complex, produced an insufficient ATP phosphorylating capacity, elevated the mitochondrial membrane potential, and caused unexpected c-subunit accumulation [27]. In addition, ATP5E was proven to be required for normal spindle orientation during embryonic divisions in *Drosophila* [28].

To date, the roles of *ATP5E* in CRC tumor development and disease progression remain unclear. Therefore, we investigated the relationship among ATP5E expression, disease stage, and survival in CRC patients. Moreover, we also investigated functional consequences of *ATP5E* alterations in CRC tumor cells, and the signaling axis that causes the EMT.

## 2. Experimental Section

### 2.1. Reagents, Cell Lines, and Lentiviral Transduction

Human CRC HCT116 and H3347 cells were obtained from American Type Culture Collection (Manassas, VA, USA) and grown in Dulbecco’s modified Eagle medium (DMEM) and RPMI medium supplemented with 10% fetal bovine serum (FBS). As previously described, stable knockdown clones of HCT116 and H3347 cells were generated using short hairpin (sh)RNA directed against the *ATP5E* gene constructed in a pGIPZ-puro vector obtained from OpenBiosystems (Huntsville, AL, USA) [29]. A plasmid carrying a non-silencing (NS) control sequence was used to create control cells. Puromycin for stable clone selection was purchased from Sigma-Aldrich (St. Louis, MO, USA).

### 2.2. CRC Sample Selection and Immunohistochemical Analysis

The studied tissues were retrieved from the Department of Pathology, Taipei Municipal Wan Fang Hospital (Taipei, Taiwan) with Institutional Review Board approval (TMU-IRB 99049). Surgical specimens had been fixed in 10% buffered neutral formalin and embedded in paraffin. The histological diagnosis, tumor size, tumor invasiveness, and lymph node status of all cases were reviewed and confirmed by two pathologists (CLF and CLC). The final disease stages were determined according to the Cancer Staging System of the American Joint Committee of Cancer (AJCC). Clinical data, including the follow-up period, overall survival period, and disease-free survival period, were retrospectively collected from the medical record of each patient. Patients were followed-up for up to 152 months. Patients who died of postoperative complications within 30 days after surgery were excluded from the survival analysis.

A tissue microarray (TMA) was used for the IHC analysis of ATP5E expression in this study. A TMA containing CRC tissues and corresponding adjacent non-cancerous colon tissues was prepared, as described previously [30]. Three 1 mm cores from different areas of a tumor tissue in a paraffin block containing the tumor were selected in each case. If available, two 1-mm cores of adjacent non-cancerous normal colon mucosa were also selected in each case. In total, 243 archival CRC samples were assembled in the TMA. Antibodies used for IHC staining included anti-human ATP5E (1:100) (Abnova, Taipei, Taiwan), p-AMPKα thr-172 (1:50) (Cell Signaling Technology, Danvers, MA, USA), and E-cadherin (1:200) (BD Biosciences, Piscataway, NJ, USA). Immuno-detection was performed with an EnVision dual-link system-horseradish peroxidase (HRP) detection kit (DAKO, Glostrup, Denmark).

A four-point staining-intensity scoring system was devised to determine ATP5E expression in CRC TMA specimens, and staining-intensity scores ranged from 0 (no expression) to 3 (high expression). The results were classified into two groups according to the intensity and extent of staining: In the low-expression group, either no staining was present (staining intensity score = 0) or positive staining was detected in fewer than 10% of cells (staining intensity score = 1); and in the high-expression group, positive immunostaining was present in 10%–30% (staining intensity score = 2) or more than 30% of cells (staining intensity score = 3). All of the IHC staining results were reviewed and independently scored by two pathologists.

### 2.3. Animal Study

All animal work was conducted in accordance with a protocol approved by the Academia Sinica Institutional Animal Care and Utilization Committee. Age-matched severe combined immune-deficiency mutation and interleukin-2 receptor gamma chain deficiency (NOD SCID gamma) female mice (6–8 weeks old) originally from Jackson Laboratory (Farmington, CT, USA) were used. For the experimental tumorigenesis assay, 5 × 10^6^ HCT 116 cells were re-suspended in 0.1 mL of phosphate-buffered saline (PBS) and subcutaneously injected into the backs of SCID mice. Tumor volumes were measured every week. If tumor masses occurred, they were harvested at the end of week 3. For the experimental metastasis assays, 10^6^ HCT 116 cells were re-suspended in 0.1 mL of PBS and injected into the lateral tail vein of SCID mice. Mouse lungs were harvested at 2.5 weeks after the injection. The number of lung metastatic nodules was measured, and the intensity of green fluorescence was quantified using a noninvasive bioluminescence system (IVIS-Spectrum, PerkinElmer, MA, USA). Tissues were fixed in 10% buffered neutral formalin and embedded in paraffin. Sections of 4 μm in thickness were stained with hematoxylin and eosin (H&E) for the histopathological analysis.

### 2.4. Western Blot Analysis

A Western blot analysis was performed with the primary antibodies anti-ATP5E (1:1000) (cat. no.: H00000514-M01, Abnova, Taipei, Taiwan), anti-pAkt ser 473 (1:1000) (cat. no.: 4060, Cell Signaling Technology), anti-Akt (1:2000) (cat. no.: 4691, Cell Signaling Technology), anti-pGSK3β (1:1000) (cat. no.: 9336, Cell Signaling Technology), anti-GSK3β (1:1000) (cat. no.: 9332, Cell Signaling Technology), anti-Snail (1:1000) (cat. no.: 3879, Cell Signaling Technology), anti-E-cadherin (1:1000) (cat. no.: 610182, BD Biosciences), and anti-α-tubulin (1:10^4^) (Sigma-Aldrich, St. Louis, MO, USA).

### 2.5. Measurements of Intracellular ATP Levels

HCT116 and H3347 cells were cultured as described above before being trypsinized and washed twice with PBS for the ATP analysis. Cells (8 × 10^4^) were applied to each well of white-wall 96-well plates (Greiner Bio One, Frickenhausen, Germany). Intracellular ATP levels were quantified using a CellTiterGlo Assay (Promega, Madison, WI, USA) and a Victor3 plate reader (PerkinElmer Life Science, Waltham, MA, USA) following the manufacturer’s instructions.

### 2.6. Invasion and Migration Analyses

For the invasion assay, polycarbonate filters were pre-coated with human fibronectin on the lower side and Matrigel on the upper side. Medium containing 10% FBS was added to each well of the lower compartment of the chamber. Cells were re-suspended in serum-free medium containing 0.1% bovine serum albumin and added to each well of the upper compartment. Cells were incubated for 16 h at 37 °C in 5% CO_2_. At the end of incubation, cells were counted under a light microscope (200×, ten random fields from each well). All experiments were performed in quadruplicate. For the migration assay, wounds were created in confluent cells using a pipette tip and then rinsed with PBS to remove any free-floating cells and debris. Wound healing was measured at 0, 12, 24, and 36 h under a light microscope (100×).

### 2.7. Statistical Analysis

All observations were confirmed by at least three independent experiments. Data are presented as the mean ± standard deviation (SD). An analysis of variance (ANOVA) was used to evaluate the statistical significance of mean values. A Cox proportional hazards regression was used to test the prognostic significance of factors in univariate and multivariate models. The Kaplan-Meier method was used for the survival analysis. All statistical tests were two-sided, and *p* < 0.05 was considered significant.

## 3. Results

This section may be divided by subheadings. It should provide a concise and precise description of the experimental results, their interpretation, as well as the experimental conclusions that can be drawn.

### 3.1. Overexpression of ATP5E in CRC Is Associated with Distal Metastasis

We first analyzed expressions of ATP synthase subunits in the GSE23878 dataset containing 36 CRC tissues and 24 non-cancerous colon tissues. A hierarchical clustering analysis showed that expression profiles of genes coding ATP synthase subunits were suppressed in tumor tissues compared to normal tissues (Figure 1a). Interestingly, we found that *ATP5E,* which encodes the ATP synthase epsilon subunit, was uniquely overexpressed in tumor tissues compared to normal tissues (Figure 1b). Next, we analyzed *ATP5E* expression in another microarray dataset from GSE41258 containing a panel of CRC samples that progressed from normal colon to polyps, primary tumors, and metastatic tumors. Surprisingly, *ATP5E* expression was significantly upregulated in primary tumors and increasingly upregulated in liver metastasis and lung metastasis (Figure 1c). To validate our findings, we performed an RT-PCR to detect mRNA expression levels of *ATP5E* in normal versus tumor tissues. In eight of nine CRC samples (88%), its expression in the tumor portion was markedly higher than that of the normal part (Figure 1d). We also performed an IHC analysis to examine ATP5E expression in 60 NT-paired CRC specimens. Staining results revealed a significant trend of higher expression of ATP5E in tumor tissues compared to normal tissues (Figure 1e,f, *p* < 0.01). To further determine the prognostic role of ATP5E, we performed an IHC analysis of 243 CRC patients with known clinical follow-up information. Among these patients (with median follow-up of 70 months for censored patients), there were 127 deaths, and demographic information is shown in Appendix A. Figure 1g illustrates representative scores for quantitating ATP5E expression based on its staining intensity. The Kaplan-Meier survival analysis showed that high levels of ATP5E were significantly correlated with worse overall survival and disease-free survival (Figure 1h,i, *p* < 0.01). Relationships of ATP5E expression levels with clinicopathological characteristics of CRC are summarized in Appendix A. Furthermore, univariate and multivariate COX regression analyses, including ATP5E scores, the tumor status, lymph node involvement, metastasis, stage, and recurrence, showed that ATP5E is indeed an independent marker of a poor prognosis in CRC patients (Appendix A).

### 3.2. ATP5E Regulates Migration and Invasion In Vitro and In Vivo

Based on the finding that ATP5E expression is associated with distant metastasis, we then hypothesized that ATP5E expression may affect the invasiveness of colon cancer cells. Figure 2a shows the endogenous expression of ATP5E in six colon cancer cell lines. To test whether ATP5E can modulate the invasive/migratory abilities of colon cancer cells, we silenced ATP5E expression in HCT116 and H3347 cells using a shRNA lentivirus (Figure 2b). A wound-healing assay showed that knockdown of *ATP5E* resulted in a ~40% reduction of the migratory ability of HCT116 and H3347 cells (Figure 2c). In addition, knockdown of ATP5E also diminished the migratory and invasive abilities of both cell lines as evaluated by a Boyden chamber assay (Figure 2d,e). To evaluate the effects of ATP5E expression on tumor metastasis in vivo, we intravenously injected HCT116 NS control cells and *shATP5E* cells into NOD-SCID mice. As shown in Figure 2f, the number of lung tumor nodules in the *shATP5E* group was 2.5-times lower than that in NS control mice (*p* = 0.0002). Furthermore, fluorescence microscopy and photon counts also displayed significant differences between the NS control and *shATP5E* mice (Figure 2g). A histopathological examination showed further evidence of decreased distant metastases of HCT116 *shATP5E* cells compared to HCT116 NS control cells (Figure 2h).

### 3.3. ATP5E Expression Induces the EMT

Since repression of ATP5E inhibited cancer cell migration and invasion *in vitro* and *in vivo*, we further investigated the possible mechanism that regulates CRC cell motility and invasiveness. According to the literature, metastatic cancer cells often acquire a mesenchymal phenotype through the EMT. To test whether ATP5E expression can modulate the EMT, we performed a Western blot analysis to detect E-cadherin and Snail expressions upon ATP5E suppression or ATP5E overexpression. Figure 3a shows that Snail expression decreased with concurrent induction of E-cadherin expression upon ATP5E silencing. Complementarily, overexpression of ATP5E in CX-1 cells upregulated Snail and resulted in E-cadherin suppression (Figure 3a). The IHC analysis in serial sections of CRC specimens also showed this inverse correlation between ATP5E and E-cadherin (Figure 3b). To confirm mRNA expression patterns of ATP5E and E-cadherin during CRC progression, we analyzed expressions of both genes from normal colon specimens to polyps, primary tumors, liver metastatic tumors, and lung metastatic tumors in the GSE41258 dataset. Interestingly, E-cadherin expression was frequently downregulated with concurrent upregulation of ATP5E in metastatic tumors compared to primary tumors (Figure 3c).

### 3.4. ATP5E Upregulation Connects the AMPK-AKT-HIF1a Signaling Axis to the EMT Phenotype in Lung Metastatic Tumors

To elucidate the possible signaling pathways for EMT induction, we extracted differentially expressed genes in lung metastatic tumors, which were predominately expression patterns of ATP5E high/E-cadherin low, from the GSE41258 dataset and subjected them to an IPA Upstream Regulator Analysis. With this approach, we found that the AKT-HIF1-α signaling axis was predicted to be activated (Figure 4a). Moreover, downstream targets of AKT and HIF1α, including fibronectin and E-cadherin, were differentially regulated (Figure 4b). While AMPK was not predicted to be activated but based on the negative regulatory relationship between AMPK and AKT, we hypothesized that phosphorylation of AMPK at thr-172 would be inhibited. To test this hypothesis, we performed a Western blot analysis to detect the phosphorylation status of AMPK and AKT. Data showed that AMPK was activated with concurrent inhibition of AKT activity upon ATP5E knockdown in HCT116 and H3347 cells (Figure 4c). Moreover, ATP5E knockdown of HCT116 cells abolished stabilization of the HIF1α protein in hypoxia (Figure 4d).

## 4. Discussion

In the present study, we determined the prognostic role of ATP5E expression in CRC patients and the functional consequences of ATP5E in two colon cancer cell lines *in vitro* and *in vivo*. To our knowledge, no previous investigation has determined the prognostic role of the *ATP5E* gene in human CRC. We concluded that the ATP5E expression status was significantly associated with both disease-free survival and overall survival. It was also inversely correlated with p-AMPK and E-cadherin expression statuses in terms of patient survival.

The role of F0/F1-ATP synthase in human cancer was evaluated in recent studies [7,16,17,31,32]. Nevertheless, most studies demonstrated that expression of the b-subunit of F0F1-ATP synthase is repressed in human cancer cells of the liver, colon, kidneys, lungs, breast, stomach, and esophagus compared to their corresponding normal tissues, whereas proteins (genes) involved in glycolysis are upregulated in most human tumors [16,31]. On the basis of these findings, it was hypothesized that the metabolic phenotype of tumor cells shifts from oxidative phosphorylation to glycolysis. However, results were obtained from a limited number of normal or tumor samples. Originally, Warburg hypothesized that cancer cells develop a defect in mitochondria that leads to impaired aerobic respiration and a subsequent strain on glycolytic metabolism, and these were supported by a number of reports [7,15,16,31,32]. However, subsequent work showed that mitochondrial function is not impaired in most cancers [33,34]. In addition, Shin et al. reported that downregulation of F0F1-ATP synthase induces an increase in 5-fluororuracil resistance [35]. Notwithstanding, F0F1-ATP synthase is overexpressed in liver metastasis of CRC, and higher expression of the b-subunit in breast cancer is related to poor outcomes for those patients [17,32]. In addition, ATP5A expression is upregulated in metastasized tumors and liver metastasis compared to primary tumors and normal cells in the colon [36]. High expression levels of ATP synthase 6 and the d-subunit of F0F1-ATP synthase were also, respectively, found in tumor samples of thyroid papillary carcinomas and lung adenocarcinomas [37]. Overexpression of the a- and b-subunits of F0F1-ATP synthase were correlated with metastasis in melanoma cell lines and lung and lymph node metastases related to primary tumors [38]. The major interpretation of these results is the direct impacts of increases in ATP synthase subunits on cellular energy transduction, which may obscure an extra contribution to the apoptotic potential resulting from the increase in mitochondrial oxidative phosphorylation. To accomplish the multistep cascade of metastasis, tumor cells may need an active supply of energy. Additionally, tumor cells have high levels of ATP and ATP synthase for their energy sources.

The role of F1F0-ATP synthase in cancer progression or metastasis has not yet been well characterized. In previous reports on clinical cancer samples, decreased expression of the b-subunit in colon and lung cancers was correlated with a poorer prognosis [7,16]. Data were reported in limited samples of the early stage, pT1+pT2, lung adenocarcinomas, and Duke’s stage B2+B3 colon adenocarcinomas. Those results were contrary to our findings. Our results demonstrated that high expression of ATP5E was correlated with a poor prognosis in our pool of all subjects. Also, highly significant differences in overall survival and disease-free survival appeared between subjects in AJCC stages 3+4 and in stages 1+2 with higher ATP5E expression and those with lower ATP5E expression (Appendix A). Our findings concurred with the concept that cancer cells require more energy to trigger metastasis. Especially, Eukaryotic translation initiation factor 4E–binding protein 1 (4E-BP1) is a key downstream effector of mTOR complex 1 (mTORC1), which regulates mitochondrial activity and improves metabolic homeostasis [39,40]. Therefore, we observed the phosphorylation status of 4E-BP1 has been reduced in ATP5E knockdown stable cells (Appendix A). Recent studies focused on ATP and ATP synthase as targets for anticancer therapies in animal and cell line models [19,20]. Combined all evidences, we hypothesized that ATP5E dysfunction in colon tumorigenesis is correlated with ATP production, mitochondrial function, and then interacts with several oncogenic pathways

AMPK is a member of a protein kinase family that is activated during energy deficiencies in order to restore ATP levels [12]. Increasing the AMP/ATP ratio induces phosphorylation of AMPK by LKB1. However, p-AMPK expression (negative or positive) is not associated with survival in CRC. Only after combining with the p-MAPK3/1 status did the prognostic effect of p-AMPK significantly differ. Notably, p-AMPK expression is associated with superior CRC-specific survival among p-MAPK3/1-positive cases [37]. According to our findings, the high expression level of p-AMPK was significantly associated with good overall survival (*p* = 0.041) and disease-free survival (*p* = 0.049) (Appendix A). The Akt signaling pathway was also interfered with by activated AMPK with a direct interaction, which might result in a reduction of glycolysis through decreases in both hexokinase activity and transcription of glycolytic enzymes [13]. Since activation of the Akt pathway was implicated in induction of the EMT, we hypothesized that silencing of *ATP5E* and increasing the AMP/ATP ratio should induce activation of AMPK to inhibit Akt, which resulted in downregulation of Snail and subsequent upregulation of E-cadherin (Appendix A) [14]. Accordingly, we found that reducing expression of the *ATP5E* gene induced activation of AMPK, and thereby inhibited Akt and GSK3β phosphorylation. This signaling gave rise to decreased stability of Snail and subsequently increased the E-cadherin expression level (Figure 4B). In addition, patients with low ATP5E expression and high E-cadherin expression had better survival than those with high ATP5E expression and low E-cadherin expression. These results suggested that ATP5E is crucial for CRC prognosis.

Many cancer cells maintain a high level of anaerobic carbon metabolism in the presence of oxygen, which is a manifestation of the Warburg effect [15]. Macrolide inhibitors, such as oligomycin, of mitochondrial F0F1-ATP synthase selectively kill metabolically active tumor cells that do not fit in the Warburg effect phenomenon. Oligomycin A has also been used to inhibit mitochondrial F0F1-ATP synthase in cancer metabolism research. Taken together with our results, the phenomenon of the Warburg effect still remains to be established in detail.

F0F1-ATP synthase (F0F1) synthesizes ATP in mitochondria coupled with proton flow driven by the Protonmotive force (PMF) across membranes. Based on previously studies, ATPase inhibitory factor 1 (ATPIF1, IF1) inhibits ATPase activity of mitochondrial F0F1-ATP synthase [41]. Under aerobic conditions, ATPIF1 make ATP from ADP and phosphate using a PMF generated by respiration, as a source of energy to drive their rotary mechanism. On the other hand, IF1-deficinet cells can maintain ATP after PMF loss by glycolysis. Therefore, we screened the expression between ATPIF1 and ATP5E in the TCGA clinical cohort (TCGA_COAD). The results showed that ATP5E was highly expressed in tumor part compared with normal adjacent tissues. In contrast, ATPIF1 is reduced in the tumor part than in the normal group. ATPIF1 and ATP5E form a significant negative correlation in clinical patients (Appendix A). However, extrinsic conditions influence (pH, ion concentration, etc.) the self-association and structure of IF1 [42]. In further experimental design, we will evaluate the mitochondrial membrane potential (△ Ψm) and identify the detailed interplay between ATP5E and ATPIF1.

## 5. Conclusions

We showed that a high ATP5E expression level was associated with a poor prognosis, including disease recurrence, overall survival, and disease-free survival in CRC. Additionally, we used CRC cell lines to investigate the roles of ATP5E in tumor growth and metastasis both in vitro and in vivo. These findings point out the potential therapeutic implications, as they indicate that mitochondrial ATP synthase inhibitors may enhance the anticancer efficacy of metabolic drugs.

## Figures and Tables

**Figure 1 jcm-08-01070-f001:**
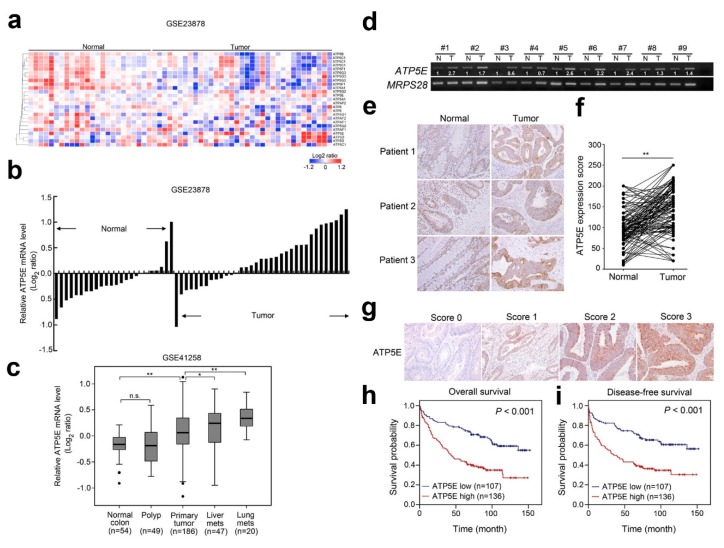
Overexpression of the ATP synthase epsilon subunit (ATP5E) in colorectal cancer (CRC) is associated with distal metastasis and a poor prognosis. (**a**) Microarray expression patterns of genes encode for ATP synthase in the GSE23878 dataset containing 36 CRC tissues and 24 non-cancerous colorectal tissues. (**b**) Relative expression of the *ATP5E* gene in the GSE23878 dataset ranked from lowest to highest. (**c**) Microarray expression patterns of the *ATP5E* gene were compared among 54 normal colon tissues, 49 polyp tissues, 186 primary tumors, 20 lung metastatic tumors, and 47 liver metastatic tumors in the GSE41258 dataset. (**d**) RT-PCR analysis of *ATP5E* levels in normal colon tissues (N) and tumor tissues (T) derived from nine patients. Data were normalized to the corresponding *MRPS28* level. (**e**) Representative IHC staining of ATP5E levels in normal colon and primary CRC tissues. (**f**) Distribution of immunoreactivity scores in normal colon and primary CRC tissues (*n* = 60). The scores were determined by the staining intensity x percentage of positive cells. (**g**) Representative scores for ATP5E IHC staining in CRC patients. (**h**) Kaplan-Meier plot of overall survival for 243 CRC patients, stratified by the ATP5E level. (**i**) Kaplan-Meier plot of disease-free survival for 243 CRC patients, stratified by the ATP5E level.

**Figure 2 jcm-08-01070-f002:**
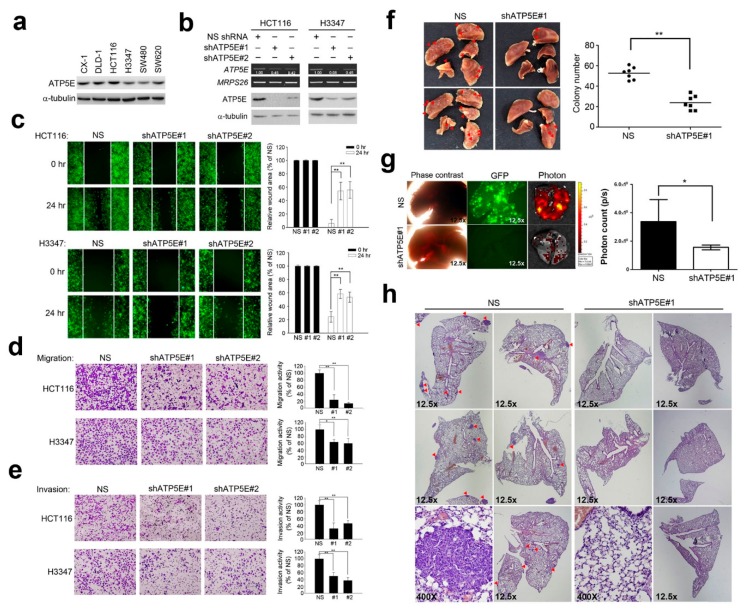
ATP synthase epsilon subunit (ATP5E) silencing inhibited invasion and migration *in vitro* and distal metastasis in vivo. (**a**) Endogenous ATP5E protein expression in six colorectal cancer cell lines. (**b**) Knockdown of ATP5E expression in HCT116 and H3347 cells by ATP5E shRNAs. The knockdown efficiency was determined by an RT-PCR and Western blot analyses. (**c**) Wound-healing assay carried out on HCT116 and H3347 cells. Relative wounded areas were compared between the non-silencing (NS) control and *shATP5E* cells at 24 h. (**d**) Migration assay for the NS control and *shATP5E* cells of the HCT116 and H3347 cell lines using Boyden chambers. (**e**) Invasion assay for the NS control and *shATP5E* cells of the HCT116 and H3347 cell lines using Boyden chambers pre-coated with Matrigel shown in the lower panel. (**f**) Representative lung images of mice injected with the NS control and *shATP5E* cells are shown in the left panel. Total numbers of lung metastatic nodules in individual mice 2.5 weeks after a tail vein injection of HCT116 NS control or *shATP5E* cells are shown in the right panel. (**g**) Green fluorescence and photon images of the lungs of mice injected with HCT116 NS control or *shATP5E* cells. The color bar represents the fluorescence intensity. (**h**) Representative H&E staining of lung sections at 12.5× and 400×. Red arrows indicate metastatic nodules.

**Figure 3 jcm-08-01070-f003:**
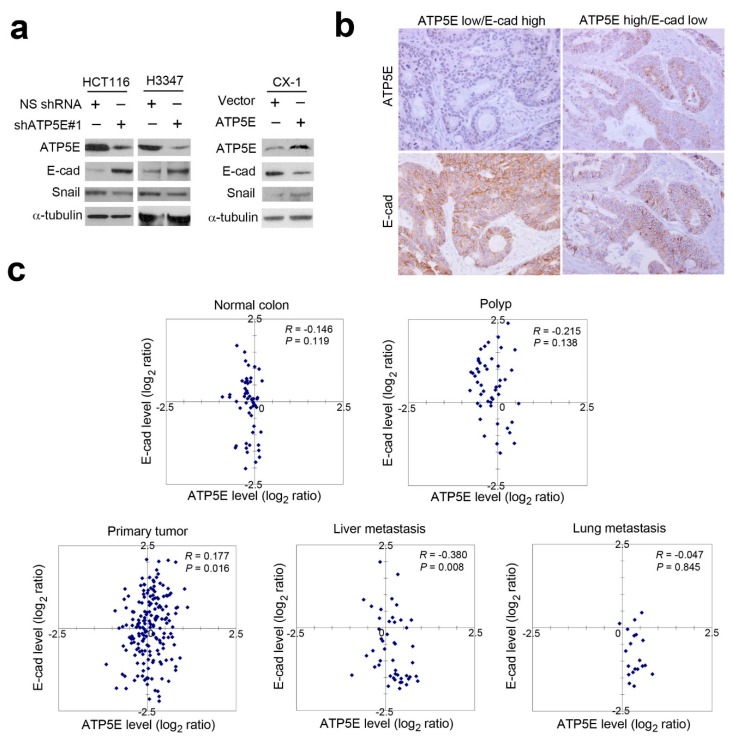
ATP synthase epsilon subunit (ATP5E) expression induces the epithelial-to-mesenchymal transition. (**a**) Western blot analysis of E-cadherin and Snail expressions upon ATP5E knockdown and overexpression. (**b**) IHC staining of ATP5E and E-cadherin in serial sections of colon cancer specimens. (**c**) ATP5E and E-cadherin expression profiles of normal colon, polyp, primary colon tumor, liver metastatic tumor, and lung metastatic tumor tissues in the GSE41258 microarray dataset.

**Figure 4 jcm-08-01070-f004:**
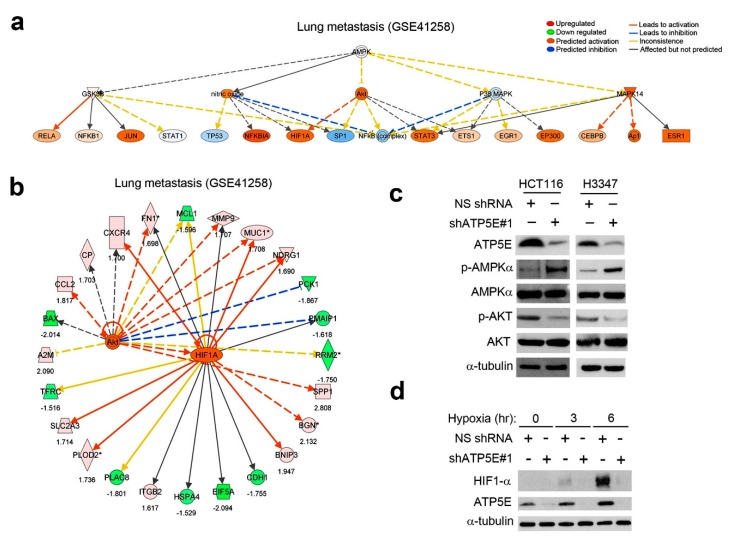
ATP synthase epsilon subunit (ATP5E) upregulation connects the adenosine monophosphate-activated protein kinase (AMPK)-AKT-hypoxia-inducible factor-1α (HIF1a) signaling axis to the epithelial-to-mesenchymal transition (EMT) phenotype in lung metastatic tumors. (**a**) Upstream regulator analysis of differentially expressed genes in lung metastatic tumors from the GSE41258 dataset. The orange circle indicates predicted activation, while the blue circle indicates predicted inhibition. (**b**) Differentially expressed genes downstream of AKT and HIF1a extracted from the GSE41258 dataset. (**c**) Western blot analysis of the phosphorylation status of AMPK and AKT upon ATP5E inhibition in HCT116 and H3347 cells. (**d**) Western blot analysis of the HIF1a protein upon ATP5E inhibition in hypoxia.

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
