# Peer review of "ATP Synthase Subunit Epsilon Overexpression Promotes Metastasis by Modulating AMPK Signaling to Induce Epithelial-to-Mesenchymal Transition and Is a Poor Prognostic Marker in Colorectal Cancer Patients"

_jcm, 2019, doi:10.3390/jcm8071070_

Round 1
Reviewer 1 Report
Authors have presented clear evidence for ATP synthase subunit epsilon in colorectal cancer metastasis. Methods and design of experiments are appropriate. Conclusions based on the results are warranted. The manuscript will attract a significant number of scientific readers specially in cancer metastasis.
Reviewer 2 Report
The article is well written and include an extensive amount of work. The finding that ATP synthase is involved in metastatic potential is not novel, and there are some conflicting evidence in the cohort reported that only the ATP5E is upregulated,verses down regulation of the other subunits of ATPsynthase. It would be interesting to see a speculation to why this is found. Furthermore, it would strengthen the article if the authors had included some functional analysis of the mitochondria in the two cell lines used to silence ATP5E. The supplementary A1 figure where ATP levels have been measured shows a significant (statistically, although not profound) decrease in the shATP5E-1 and shATP5E-2, which would be further strengthened if one could also show a relation to functional mitochondrial studies in these cells.
It would also be interesting to hear how the authors arrived at their ATP5E scoring system, where less than 10% was 1, above 10-30% was given score 2 and anything above 30% was score 3. These cut off values are used throughout the following analysis and impact the derived conclusions, which makes it an important point to elaborate on.
The discussion could include some more speculation to the role of ATP5E in ATP synthesis compared to the other subunits. An interesting hypothesis has previously been suggested by Fenouik et. al (ref. doi:10.1016/j.febslet.2005.08.030), where membrane potential plays a role in how ATPepsilon regulate ATP synthesis vs hydrolysis.
